# HVCR: Causal Evaluation of Large Multimodal Models in Human-like Video Reasoning

## Abstract

The pursuit of human-like causal reasoning in large multimodal models (LMMs) is a critical yet challenging frontier. Current video causal reasoning benchmarks often lack a systematic design that aligns with the nuances of human causal cognition. To address this gap, we introduce the HVCR benchmark for evaluating human-like video causal reasoning in LMMs, systematically designed across three levels. At the **definitional** level, since causal relations in videos are inherently *specific* and naturally align with the field of actual causality, we adopt definitions from this domain that robustly handle complex scenarios like preemption, which can not be modeled by the simple "but-for" test. At the **goal-oriented** level, our aim is to simulate human causal judgments rather than fitting formal definitions or frameworks. Therefore, we establish our gold standard using human "consensus" from rigorous human experiments in cognitive science, leveraging seven well-studied causal scenarios as reliable references. At the **representational level**, we employ explicit *causal graphs* and a variant of *twin networks* to enable automatic generation of causal questions. The HVCR benchmark contains 300 videos (240 synthetic and 60 realistic) and 4,967 causal questions. These questions span three causal rungs (discovery, intervention, and counterfactual) and eight types, focusing on key aspects of human-like causal reasoning such as causal attribution and responsibility. Human evaluation shows that average observers achieve nearly 80% accuracy on our synthetic videos, confirming their clarity. However, current LMMs underperform on both synthetic and real-world videos, revealing a significant gap in their human-like causal reasoning capabilities. To our knowledge, HVCR is the first video causal reasoning benchmark to systematically integrate these three design levels, jointly consider synthetic and real-world settings, and focus exclusively on pure causal reasoning of LMMs.

## 1 Introduction

> *Where causation is concerned, a grain of wise subjectivity tells us more about the real world than any amount of objectivity.*
>
> — *Pearl & Mackenzie (2018)*

The pursuit of human-like causal reasoning is a critical frontier for AI (Chi et al., 2024; Joshi et al., 2024; Hong et al., 2024). However, current progress in video causal reasoning lacks a systematic design that simulates human causal judgments from three levels.

**Definitional Level.** Causal relationships in videos are inherently *specific* rather than *general*. For example, in a video where Alice spills water onto a rice cooker, causing it to short-circuit and catch fire, one can infer the *specific* causal relation "Alice's action caused the rice cooker to catch fire." In contrast, the *general* claim "spilling water on an appliance causes short-circuit" cannot be justified from a single event. This shows that video causality naturally aligns with *specific* actual causality (Halpern, 2016), unlike textual scenarios where both general and specific causal relations coexist. Therefore, video causality should be modeled with techniques from actual causality. Nevertheless, much existing research do not specify definitions or still relies on the "but-for" test from legal theory, where $X$ is a cause of $Y$ if, but for $X$, $Y$ would not have occurred (Yi et al., 2020; Foss et al., 2025). While simple and general, this test often fails to capture human intuition. A classic case is late

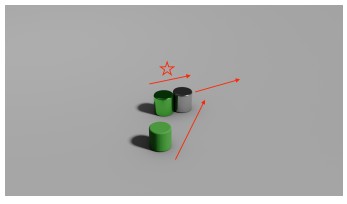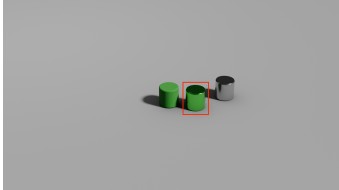

Figure 1: An example of Late Preemption.

preemption (Lewis, 2000): *Suzy and Billy both throw rocks at a bottle. Suzy's rock arrives first and shatters it. Since both throws are perfectly accurate, Billy's would have shattered the bottle had it not been preempted by Suzy's.* This case can be readily mapped into a video, as shown in Figure 1. Here, the but-for test incorrectly concludes that Suzy's throw is not a cause. This underscores the need for more precise definitions from actual causality, such as the Halpern-Pearl definition (Halpern, 2016), when modeling causal relations in videos.

**Goal-Oriented Level.** Beyond definitional foundations, the goal of causal reasoning in LMMs should be *simulating human causal cognition* rather than strictly *fitting a formal definition or framework*. Although theories such as Force Dynamics (Talmy, 1988; Wolff, 2007), Mental Models (Khemlani et al., 2014), Causal Models (Goldvarg & Johnson-Laird, 2001; Sloman & Sloman, 2009), and Counterfactual Simulation (Halpern & Pearl, 2005; Gerstenberg et al., 2021) aim to capture human causal cognition, their predictions often diverge from actual human judgments, especially in complex physical events (Mao et al., 2022). Large-scale crowdsourced annotation is one approach (Foss et al., 2025), but carefully controlled experiments, such as randomized controlled trials (RCTs) in psychology, philosophy, and cognitive science, provide more reliable references (Cartwright, 2010). These results serve as a "consensus" of human perception, reducing reliance on massive annotation while grounding models in empirically validated human reasoning.

**Representational Level.** Explicit causal representations are essential for video-based causal reasoning benchmarks. Most existing work omits explicit representations of causal relationships (Yi et al., 2020; Foss et al., 2025), while others do (Chen et al., 2024b; 2025; Yi et al., 2020).

In this work, we propose the HVCR benchmark for evaluating human-like video causal reasoning in LMMs, systematically designed across three levels. At the **definitional level**, we adopt definitions of actual causality as our foundation, which not only guides the selection of causal scenarios but also serves as a formal reference for comparison with human causal judgments. At the **goal-oriented level**, our aim is to *simulate human causal cognition*. To this end, we select seven well-studied causal scenarios that provide well-defined causal structures and human "consensus" gold standards (Kueffner, 2021). At the **representational level**, we employ *causal graphs* to explicitly encode general causal relationships, and a variant of *twin networks* to represent factual and counterfactual worlds, which are then used to automatically generate causal questions and answers.

The resulting HVCR benchmark contains 300 videos and 4,967 causal questions. It comprises 240 synthetic videos generated from six scenarios, primarily involving object motion and interactions on an infinite horizontal tabletop, and 60 real-world videos covering sports events and traffic accidents. The questions span three causal rungs (Discovery, Intervention, and Counterfactual) and eight types, covering key aspects of human-like causal reasoning such as causal attribution and responsibility assignment. Human evaluation demonstrates that average observers can accurately understand the causal dynamics in synthetic videos, achieving nearly 80% accuracy. In contrast, current LMMs underperform on both synthetic and real-world videos. To our knowledge, HVCR is the first video causal reasoning benchmark systematically designed across three levels, jointly considers both synthetic and real-world settings, and focuses exclusively on evaluating pure causal reasoning abilities in LMMs.

## 2 PRELIMINARIES

### 2.1 DEFINITIONAL LEVEL: ACTUAL CAUSALITY

Causality research in computer science can be divided into two subfields (Kiciman et al., 2024). *Type causality* concerns general causal relationships between variables, e.g., "smoking causes lung

cancer," while *actual causality* (or token causality) identifies the causes of specific events and assigns responsibility, e.g., "David's 30 years of smoking caused his lung cancer" (Halpern, 2016; Hausman, 2005). The aim of actual causality is to **axiomatize the intuitive form of causal reasoning employed by humans** (Kueffner, 2021), supporting applications such as legal reasoning, machine failure debugging, root cause analysis, and explainable AI (Lagnado & Gerstenberg, 2017; Andreas et al., 2023; Lu et al., 2023; Dubslaff et al., 2022; Sharma et al., 2022; Blake et al., 2025; Chockler et al., 2024).

## 2.2 REPRESENTATIONAL LEVEL: CAUSAL GRAPHICAL MODELS

**Causal graphs** are directed acyclic graphs (DAGs) that depict causal relationships between variables (Figure 2a). In this work, we use causal graphs to represent general causal relationships between events. This is an implicit encoding of different worlds.

**Twin networks** can be viewed as an extension to causal graphs, where two interlinked networks are employed to represent the factual and counterfactual worlds in one network (Balke & Pearl, 1994) (Fig-

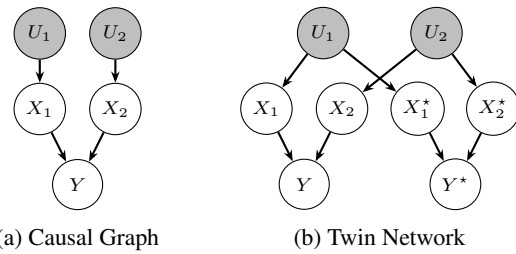

(a) Causal Graph      (b) Twin Network

Figure 2: Example of causal graphical models.

ure 2b). These two networks are identical in structure and share the same exogenous variables, as exogenous variables are not modified by interventions on observed variables. In this work, we use twin networks to represent the "states" of the factual and counterfactual worlds. That is, nodes represent assignments in both worlds, while edges have two states depicting whether the direct causal influence actually takes effect.

## 2.3 THE LADDER OF CAUSATION

The Ladder of Causation (Pearl & Mackenzie, 2018) illustrates three rungs of causal reasoning: Association, Intervention, and Counterfactual. Since association does not involve causal inference, we instead regard Discovery as Rung 1 (Chen et al., 2024a). **Discovery** (Rung 1) involves identifying cause-effect pairs from observational data without prior knowledge, e.g., "*Is there a causal relationship between review frequency and exam scores?*" **Intervention** (Rung 2) involves exploring the effects of manipulating variables using the *do*-operator, e.g., "*What if I review everyday, will my exam scores improve?*" **Counterfactual** (Rung 3) considers hypothetical alternatives, e.g. "*What if I have attended a party instead of reviewing, would my exam scores be good?*"

## 3 DATASET CONSTRUCTION

HVCR comprises 300 videos, of which 240 are synthetic and 60 are realistic. The dataset contains a total of 4967 causality-focused QA pairs. The construction pipeline for HVCR is shown in Figure 3.

## 3.1 CAUSAL SCENARIO SELECTION

The field of actual causality (and causal judgment) often identify seven real-world causal scenarios (Kueffner, 2021). These scenarios are notable because human causal judgments often conflict with formal definitions and models of actual causality. This discrepancy has driven the development of new definitions and more accurate causal models. Therefore, focusing on these specific scenarios and addressing these conflicts naturally aligns with human intuition. This paper focuses on six of these scenarios, summarized below. All six scenarios are included in the synthetic data, while the realistic data only considers SW, LP, and BP, as they are more common in real-world situations. The six scenarios are typically represented by the neuron diagrams shown in Figure 3.

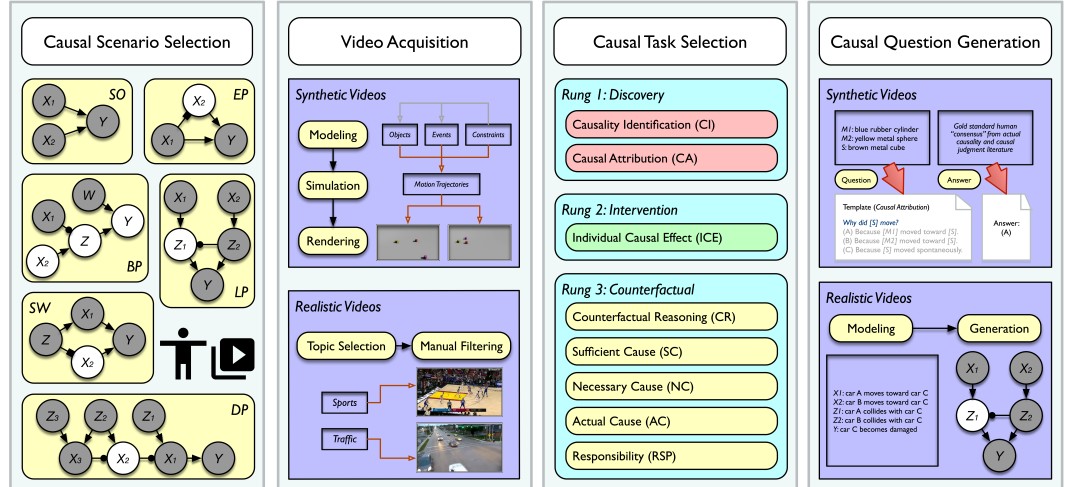

Figure 3: The Construction Pipeline of `HVCR`.

**Symmetric Overdetermination** (SO): Multiple processes, all producing the same outcome, terminate at the same time.

**Switch** (SW): An event triggers one of two processes, both of which have the same outcome, making the event immaterial for the final result.

**Late Preemption** (LP): Two causal processes run in parallel; both would produce the same outcome, but one terminates before the other, rendering the latter irrelevant.

**Early Preemption** (EP): Two causal processes could produce the same outcome, but one terminates before the other even starts.

**Double Preemption** (DP): A process that would have prevented another process is itself prevented by a different process.

**Bogus Preemption** (BP): An action is taken to interrupt a process that was never active.

## 3.2 VIDEO ACQUISITION

### 3.2.1 PRINCIPLES

We followed these principles for generating synthetic videos and collecting realistic ones. **First, "one-take" exhibition.** A single-take video should be enough to convey all the prior knowledge needed for the causal scenarios and be easily understood by the average human observers. **Second, quality control.** We focused on creating or collecting high-quality videos that adhere to the first principle. **Third, diversity design.** We aimed for multi-dimensional diversity, including variations in data realism, entities, events, topics, causal structures, and perturbed settings, etc..

### 3.2.2 SYNTHETIC DATA GENERATION

**Objects & Events.** Objects in `HVCR` follow similar compositional attributes as in `CLEVRER` (Yi et al., 2020), including three shapes (cube, sphere, and cylinder), two materials (metal and rubber), and eight colors (gray, red, blue, green, brown, cyan, purple, and yellow). In each video, identical objects are prohibited, where each combination of the three attributes uniquely identifies a single object. We consider both unary and binary events. For **unary events**, we focus on the *motion status* of an object, which shows whether the object is moving or stationary at a time point. For **binary events**, we examine: 1) *collisions* and 2) *relative movement directions*. A moving object may approach a stationary one or another moving object, indicating potential collisions.

**Modeling.** First, we define unique **events and constraints** for each scenario and control initial object attributes (e.g., location, orientation, velocity, and angular velocity) to realize these events in videos. The constraints primarily involve 1) the number and sequence of collisions and 2) the "one-take" exhibition of knowledge from both factual and counterfactual worlds. For example, in LP, we consider the moving directions of M1 and M2 relative to S, the motion status of S, and the

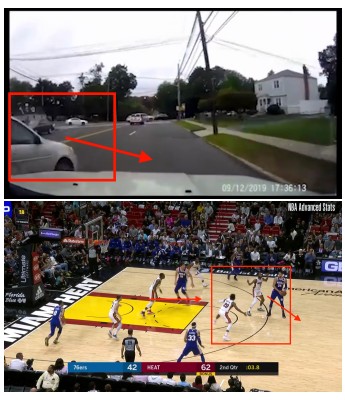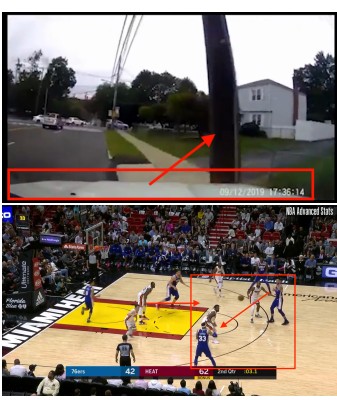

Figure 4: Top: **SW** The white car in front changes lanes suddenly, leaving the driver no time to stop. Whether staying on course or turning right, the vehicle is damaged. The driver ultimately turns right and crashes into a utility pole. Bottom: **BP** 76ers No. 25 (in blue) fakes a shot, causing HEAT No. 5 (in white) to jump, then passes to 76ers No. 33. HEAT No. 5 interrupts an inactive process.

collision between M1 and S. The number of collisions is exactly one, and M2 must be factually and counterfactually able to reach S's initial position to ensure sufficiency. Variations in object attributes increase diversity, while constraints help uphold the "one-take" principle.

**Simulation.** Second, we use the Bullet physics engine (Coumans, 2010) to simulate the objects' motion trajectories. Object attributes are implemented flexibly, with values randomized from a specified range. They are also adjusted to enable early modeling of counterfactual knowledge. For constraints, we designed *automatic filters* to remove simulations where the number and sequence of collisions did not align with our modeling.

**Rendering.** Third, we used Blender (Blender Online Community, 2016) to render the motion trajectories into realistic videos. Due to the challenges of "one-take" modeling of counterfactual knowledge and potential issues with the camera angle (such as objects being out of view or unclear collisions), we manually selected videos that adhered to the "one-take" principle.

### 3.2.3 REALISTIC DATA COLLECTION

Realistic videos are collected from existing video benchmarks that reflect at least one of our causal scenarios. Here, we focus on SW, LP, and DP, as they are common in real-world interactions.

**Topics & Sources of Videos.** We focused on two primary topics: *multi-person sports* and *traffic accidents*. Sports videos, which are excellent for capturing multi-person interactions fitting our scenarios, are collected from MultiSports (Li et al., 2021) and NSVA (Wu et al., 2022). These included Olympic and World Cup volleyball, as well as NBA basketball. Videos of traffic accidents, involving collisions between vehicles and people, were sourced from V-TIDB (Nishiyama et al., 2023), TAD (Xu et al., 2024), and TU-DAT (Pradeep Kumar & Kant, 2025). These videos included both ego-centric and camera-view perspectives. In total, we collected 20 videos for each scenario.

**Entities & Events.** The topics involve *multi-entity interactions*, including human-human (e.g., in volleyball and basketball games), human-object (e.g., vehicle-pedestrian collisions), and object-object (e.g., vehicle-vehicle collisions). Events vary by topic: for volleyball, they include *fake attacks* and *block attempts*; for basketball, they include *fake shots* and *securing jump balls*; and for traffic accidents, they involve events like *turning directions* and *collisions*. As a result, realistic videos contain a richer variety of both entities and events.

**Typical Examples.** In **SW**, a common example is a traffic accident where a vehicle in the left lane suddenly cuts in front of the following vehicle. The driver of the latter cannot slow down in time. Whether they try to brake or swerve right, the vehicle will collide with either the lane-changing car or the guardrail/another vehicle in the right lane. An example is shown in Figure 4. A typical **LP** situation in basketball involves two teammates, one behind the other, both going for a rebound. Regardless of who gets the ball, their team gains possession. In traffic, a variant of LP occurs when

an oncoming vehicle crashes into the front side of a car, and the driver cannot stop in time, resulting in a collision. This variant, where the two causal processes are not independent, is called dependent preemption (Moore, 2012). The **BP** scenario is particularly interesting and often involves feints in sports. For instance, in basketball, a player might fake a shot to trick a defender into jumping too early. In volleyball, a player might fake an attack jump, causing an opponent to misjudge and commit to a defensive block prematurely. An example is shown in Figure 4.

## 3.3 CAUSAL TASK SELECTION

**Rung 1: Discovery.** We include Causality Identification (CI, e.g., "*Does the car turning left affect whether it becomes damaged?*") and Causal Attribution (CA, e.g., "*Why did the car become damaged?*") in this rung.

**Rung 2: Intervention.** We include Individual Causal Effect (ICE, e.g., "*If we force the car to stay straight, will the car in front of it cause it to become damaged?*") in this rung.

**Rung 3: Counterfactual.** We include Counterfactual Reasoning (CR, e.g., "*If the car had not turned left, would it still have become damaged?*"), Sufficient Cause (SC, e.g., "*Was the fact that the car turned left sufficient for it to become damaged?*"), Necessary Cause (NC, e.g., "*Was the fact that the car turned left necessary for it to become damaged?*"), Actual Cause (AC, e.g., "*What is the actual cause of the car becoming damaged?*"), and Responsibility (RSP, e.g., "*Was the fact that the car turned left responsible for it becoming damaged?*") in this rung.

We distinguish tasks reflecting natural human reasoning from those representing more abstract or formal processes. "Human" tasks involve intuitive, token-level reasoning to explain a single occurrence. CA and RSP are "Human" as they align with human intuitions (Halpern & Pearl, 2005; Halpern, 2016). CR, SC, and NC are also "Human", foundational to actual causality and causal judgment (Kiciman et al., 2024). ICE is "Human" since it concerns intervening on a single unit (Pearl et al., 2016), mirroring the human cognitive process of predicting consequences through mental simulation (Sloman & Sloman, 2009). In contrast, CI is "Non-Human" as it learns a general causal rule, not interpreting a single event. Similarly, AC is "Non-Human" since it outputs formal causal models that aim to simulate human judgment but may diverge systematically (Lagnado & Channon, 2008).

## 3.4 CAUSAL QUESTION CONSTRUCTION

**Causal Structures.** In `HVCR`, we provide causal graphs and twin networks for each video. Synthetic videos have two settings: 1) Basic, which strictly follow the causal structures defined by the scenario, with events uniquely defined in the modeling stage; 2) Perturbed, which add 1-2 isolated nodes (e.g., 1 stationary object, 2 stationary objects, 1 moving object, or 2 moving objects). These objects do not interact with the main scenario objects, and the perturbations increase the options for CA and AC. For realistic videos, we manually define the entities and events. The causal structures are similar to the original scenarios but more diverse, especially for LP (e.g., dependent preemption). Entities and events also contribute to this diversity.

Table 1: Human evaluation results.

| Question Type | # Videos | # QAs | Acc. |
|---|---|---|---|
| *Human* | | | |
| Causal Attribution | 6 | 6 | 76.67% |
| Individual Causal Effect | 6 | 14 | 75.71% |
| Counterfactual Reasoning | 6 | 13 | 76.92% |
| Sufficient Cause | 6 | 15 | 76.00% |
| Necessary Cause | 6 | 15 | 78.67% |
| Responsibility | 6 | 15 | 88.00% |
| **Overall** | 36 | 78 | **78.97%** |
| *Non-Human* | | | |
| Causality Identification | 6 | 16 | 61.25% |
| Actual Cause | 6 | 6 | 56.67% |
| **Overall** | 12 | 22 | 60.00% |
| **Total** | 48 | 100 | 60.00% |

**Question Generation.** Based on the causal structure, we generate causal questions using a template-based approach. For videos where the entities, events, and causal structures match the original scenario, we use the corresponding answers directly. For videos with any changes, we manually verify the consistency of the answers.

| Rung/Type | # QAs | Avg Q/A Len |
|---|---|---|
| **R1** | | |
| CI (YN) | 804 | 21.1/1.0 |
| CA (MC) | 300 | 13.1/8.5 |
| **R2** | | |
| ICE (YN) | 700 | 30.4/1.0 |
| **R3** | | |
| CR (YN) | 640 | 23.3/1.0 |
| SC (YN) | 741 | 24.5/1.0 |
| NC (YN) | 741 | 24.5/1.0 |
| AC (MC) | 300 | 13.1/8.5 |
| RSP (YN) | 741 | 23.6/1.0 |
| **Overall** | **4967** | **22.9/2.0** |

Figure 5: Statistics of `HVCR`. YN, Yes/No; MC, Multiple Choice.

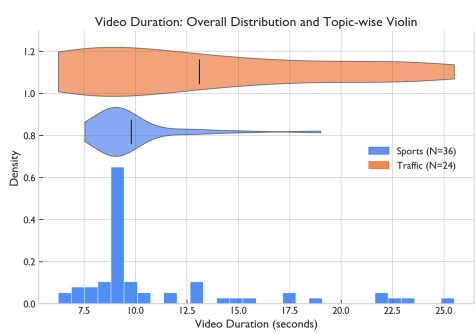

Figure 6: Duration distribution of realistic videos in `HVCR`.

## 4 BASELINE EVALUATION

### 4.1 HUMAN EVALUATION

For synthetic videos, we have conducted a human evaluation to show *whether these generated videos truly reflect our principles*. In this test, we select 8 videos for each scenario in the Basic setting, each corresponding to a question type. The resulting size of the test is 48 videos and 100 questions. Totally, 5 people on the Prolific platform participated in the test. The results are shown in Table 1. For human tasks, the average human score is 78.97%. This suggests that our synthetic "one-take" videos are basically sufficient to convey causal knowledge of both the factual and counterfactual worlds, in line with human intuition. Also, this provides a small-scale baseline for evaluation.

### 4.2 LARGE MULTIMODAL MODELS

We evaluate a range of LMMs on the `HVCR` benchmark, including: 1) **Open-source "smaller" models:** LLaVA-OneVision-Qwen2-7B (Li et al., 2025a), Qwen2.5-VL-7B/32B-Instruct (Bai et al., 2025a), InternVL2.5-8B (Chen et al., 2024c), and Perception-LM-8B (Cho et al., 2025). Each model was run on a single NVIDIA H200 (141GB) GPU on our research cluster. Models are run with 16 uniformly sampled frames. Temperatures are set to zero for reproducibility. 2) **Closed-source "larger" models:** GPT-4o (OpenAI, 2024) and Gemini-2.5-Flash (Comanici et al., 2025). Both models are evaluated using 10 frames, uniformly sampled. Again, temperatures are set to zero.

### 4.3 RESULTS & DISCUSSION

#### 4.3.1 RESULTS ON SYNTHETIC VIDEOS

**Overall Results.** As shown in Table 2, Qwen2.5-VL-32B stands out with the best overall performance, achieving 58.33% accuracy, far surpassing large, closed-source models such as GPT-4o and Gemini-2.5-Flash. Among open-source models, InternVL2.5 leads, closely followed by Perception-LM, both outperforming closed-source alternatives. In contrast, models like LLaVA-OneVision and Qwen2.5-VL-7B show poorer results, especially the latter. Upon reviewing the failure modes of Qwen2.5-VL-7B, we found that it tends to select both answers for Yes/No questions when uncertain. The scaling effect is evident, with the 32B version showing a notable performance improvement. Closed-source models all fall below 50% accuracy, suggesting their open-source counterparts promising for video causal understanding.

**Human vs. Non-Human.** Of the 7 models tested, more than half (4 models) performed worse on Human tasks than on Non-Human tasks, indicating that the models still struggle with human-like causal reasoning. Furthermore, the high performance of these models on CI and AC tasks (especially Gemini-2.5-Flash, InternVL2.5, and Perception-LM, with AC accuracy above 80%) contrasts sharply with their lower performance on tasks like CA and RSP, suggesting that the models are better at matching formal frameworks than at reflecting human causal cognition.

Table 2: Performance of different models on synthetic videos in `HVCR`.

| | | | | Closed | | Open | | | | |
| | | | | GPT-4o | Gemini-2.5-Flash | InternVL2.5 (8) | LLaVa-OneVision (7) | Perception-LM (8) | Qwen2.5-VL (7) | Qwen2.5-VL (32) |
| | | # Videos | # QAs | | | | | | | |
|---|---|---|---|---|---|---|---|---|---|---|
| *Accuracy per scenario per setting* | | | | | | | | | | |
| SO | Basic | 20 | 280 | 42.50% | 55.36% | 54.64% | 21.07% | 55.71% | 10.00% | 64.29% |
| | Perturbed | 20 | 280 | 38.93% | 47.86% | 55.36% | 21.43% | 53.57% | 7.14% | 71.43% |
| SW | Basic | 20 | 360 | 49.17% | 51.39% | 46.67% | 39.72% | 43.61% | 26.11% | 56.11% |
| | Perturbed | 20 | 360 | 46.67% | 49.17% | 48.33% | 36.94% | 51.78% | 11.67% | 53.61% |
| LP | Basic | 20 | 300 | 42.00% | 44.67% | 60.67% | 23.33% | 62.67% | 15.00% | 64.33% |
| | Perturbed | 20 | 300 | 35.33% | 44.00% | 53.33% | 20.33% | 55.33% | 11.33% | 61.33% |
| EP | Basic | 20 | 280 | 39.29% | 41.79% | 58.57% | 26.79% | 63.57% | 19.64% | 63.21% |
| | Perturbed | 20 | 280 | 28.93% | 39.64% | 51.43% | 25.00% | 62.86% | 18.57% | 58.21% |
| DP | Basic | 20 | 400 | 69.75% | 62.75% | 70.50% | 36.75% | 52.00% | 44.75% | 48.00% |
| | Perturbed | 20 | 400 | 64.50% | 65.25% | 70.00% | 34.00% | 53.75% | 40.25% | 50.75% |
| BP | Basic | 20 | 380 | 39.74% | 41.58% | 46.58% | 44.74% | 38.42% | 38.16% | 58.42% |
| | Perturbed | 20 | 380 | 35.26% | 43.68% | 51.84% | 45.79% | 39.21% | 36.05% | 58.95% |
| *Accuracy per rung per type* | | | | | | | | | | |
| R1 | CI | 240 | 640 | 63.44% | 44.22% | 56.09% | 13.44% | 66.25% | 0.78% | 40.78% |
| | CA | 240 | 240 | 49.17% | 45.83% | 47.92% | 24.58% | 51.25% | 25.42% | 41.67% |
| R2 | ICE | 240 | 560 | 27.68% | 43.93% | 32.68% | 35.71% | 36.61% | 31.07% | 67.68% |
| R3 | CR | 240 | 520 | 28.46% | 41.15% | 62.69% | 15.96% | 36.35% | 23.85% | 57.12% |
| | SC | 240 | 600 | 39.33% | 41.83% | 83.33% | 28.50% | 81.17% | 16.00% | 69.83% |
| | NC | 240 | 600 | 38.17% | 50.50% | 31.83% | 67.83% | 31.83% | 48.33% | 57.67% |
| | AC | 240 | 240 | 69.17% | 85.42% | 80.83% | 49.17% | 89.58% | 31.25% | 55.00% |
| | RSP | 240 | 600 | 60.00% | 61.50% | 61.33% | 29.00% | 56.00% | 27.83% | 66.50% |
| *Accuracy of human vs. non-human tasks* | | | | | | | | | | |
| Human | | - | - | 39.94%↓ | 47.85%↓ | 53.94%↓ | 35.06%↑ | 49.07%↓ | 29.23%↑ | 62.18%↑ |
| Non-Human | | - | - | 65.00% | 55.45% | 62.84% | 23.18% | 72.61% | 9.09% | 44.66% |
| *Aggregated accuracy* | | | | | | | | | | |
| Basic | | 120 | 2000 | 48.10% | 50.00% | 56.30% | 33.20% | 54.20% | 27.30% | 58.30% |
| Perturbed | | 120 | 2000 | 42.80% | 49.05% | 55.50% | 31.70% | 54.30% | 22.30% | 58.35% |
| **Overall** | | 240 | 4000 | 45.45% | 49.53% | 55.90% | 32.45% | 54.25% | 24.80% | 58.33% |

**Analysis by Scenario and Setting.** Models vary in their understanding of different scenarios. Apart from Qwen2.5-VL-32B, which achieves over 50% accuracy in all scenarios except DP, other models tend to perform well in certain scenarios but struggle in others. For instance, InternVL2.5 excels in understanding DP, but performs poorly in SW.

**Analysis by Rung and Type.** From the radar chart shown in Figure 7, it is evident that all models perform best and most consistently in the counterfactual rung, while there is a significant variation in performance across models in the discovery and intervention rungs. Additionally, some failure modes emerge. For instance, Qwen2.5-7B and LLaVA-OneVision primarily struggle with the discovery rung, while GPT-4o performs poorly in the intervention rung.

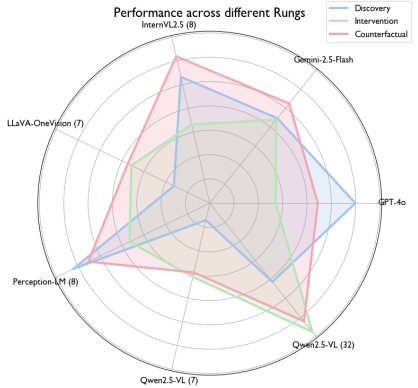

Figure 7: The performance of LMMs on synthetic videos by causal rung.

### 4.3.2 RESULTS ON REALISTIC VIDEOS

**Overall Results.** Perception-LM is the only model to surpass 50% accuracy, achieving 58.22%, followed by Gemini-2.5-Flash and Qwen2.5-VL-32B. For synthetic data, the top-performing model, InternVL2.5, now ranks at the bottom, with Qwen2.5-VL-32B also falling below 50%. This highlights the gap between synthetic and real-

Table 3: Performance of different models on realistic videos in `HVCR`.

| | | # Videos | # QAs | Closed | | Open | | | | |
| | | | | GPT-4o | Gemini-2.5-Flash | InternVL2.5 (8) | LLaVa-OneVision (7) | Perception-LM (8) | Qwen2.5-VL (7) | Qwen2.5-VL (32) |
|---|---|---|---|---|---|---|---|---|---|---|
| *Accuracy per scenario per setting* | | | | | | | | | | |
| SW | Basic | 20 | 303 | 43.56% | 51.49% | 43.89% | 41.58% | 51.16% | 40.59% | 53.47% |
| LP | Basic | 20 | 284 | 45.77% | 44.01% | 33.80% | 33.10% | 64.79% | 27.82% | 45.07% |
| BP | Basic | 20 | 380 | 39.21% | 52.89% | 38.16% | 40.26% | 58.95% | 40.79% | 48.16% |
| *Accuracy per rung per type* | | | | | | | | | | |
| R1 | CI | 60 | 164 | 62.20% | 62.80% | 30.49% | 15.85% | 79.27% | 23.78% | 74.39% |
| | CA | 60 | 60 | 23.33% | 50.00% | 43.33% | 41.67% | 38.33% | 45.00% | 26.67% |
| R2 | CDE | 60 | 140 | 53.57% | 50.00% | 45.00% | 35.71% | 55.71% | 37.14% | 55.71% |
| R3 | CR | 60 | 120 | 64.17% | 63.33% | 31.67% | 18.33% | 62.50% | 23.33% | 60.00% |
| | SC | 60 | 141 | 19.15% | 26.95% | 22.70% | 19.86% | 70.92% | 24.11% | 29.08% |
| | NC | 60 | 141 | 31.21% | 48.23% | 47.52% | 82.98% | 36.88% | 67.38% | 37.59% |
| | AC | 60 | 60 | 55.00% | 55.00% | 68.33% | 65.00% | 75.00% | 45.00% | 60.00% |
| | RSP | 60 | 141 | 27.66% | 45.39% | 40.43% | 46.81% | 42.55% | 39.01% | 39.01% |
| *Accuracy of human vs. non-human tasks* | | | | | | | | | | |
| Human | | 60 | 743 | 37.15%↓ | 46.57%↓ | 38.09%↓ | 41.45%↑ | 52.22%↓ | 39.17%↑ | 42.40%↓ |
| Other | | 60 | 224 | 60.27% | 60.71% | 40.63% | 29.02% | 78.13% | 29.46% | 70.54% |
| *Aggregated accuracy* | | | | | | | | | | |
| Overall | | 60 | 967 | 42.50% | 49.84% | 38.68% | 38.57% | 58.22% | 36.92% | 48.91% |

world evaluations, suggesting that *perception* ability might be a crucial factor in real-world scenarios.

**Human vs. Non-Human.** The gap between human and non-human tasks becomes even more pronounced in real-world situations. The number of models with lower performance on human tasks increases from 4 to 5, and the disparity is particularly significant for GPT-4o and Qwen2.5-VL-32B.

## 5 RELATED WORK

The field of video causal reasoning has seen the introduction of a variety of benchmarks designed to assess model capabilities in understanding dynamic scenes. Datasets like CLEVRER (Yi et al., 2020), CRAFT Ates et al. (2022), and IntPhys (Riochet et al., 2018) use synthetically generated videos to control for visual complexity while focusing on physical interactions and causal events. CausalVQA (Foss et al., 2025) and NExT-QA (Xiao et al., 2021) shift the focus to real-world videos, with the former specifically designed to test physical causality in egocentric settings. To challenge models with more complex, long-form videos and reasoning tasks, benchmarks such as CausalStep (Li et al., 2025b), MECD (Chen et al., 2024b), and MECD+ (Chen et al., 2025) were introduced, with MECD focusing on discovering comprehensive event-level causal graphs. Other benchmarks, including MVP-Bench (Li et al., 2024), GRASP (Jassim et al., 2024), VCRBench (Sarkar & Etemad, 2025), and Video-Holmes (Cheng et al., 2025), evaluate more fine-grained reasoning abilities like detecting shortcuts, grounding language to physical actions, or actively seeking clues in long videos.

## 6 CONCLUSION

`HVCR` introduces a novel and systematic approach to evaluating human-like causal reasoning in large multimodal models (LMMs), providing a critical tool for assessing the nuanced capabilities required for complex video causal cognition. By aligning with the principles of actual causality, incorporating human consensus in goal-oriented evaluation, and leveraging causal graphical models as representations, `HVCR` offers a comprehensive framework that extends beyond current benchmarks. Despite promising results in human evaluation, the performance gap observed in LMMs highlights the challenges these models face in replicating human-like causal judgments.

## 7    ETHICS STATEMENT

The authors of this paper have read and adhere to the ICLR Code of Ethics. Our research involves human evaluation, with labels collected through the Prolific platform. All participants were compensated for their time and provided informed consent. The study was conducted in a manner that respects the privacy and well-being of all human subjects, and no personally identifiable information was collected. We have taken care to ensure that the benchmark is created and released in a manner that avoids potential misuse or harm, aligning with our goal of responsibly advancing the field of large multimodal models.

## 8    REPRODUCIBILITY STATEMENT

To ensure the reproducibility of our work during the reviewing process, we provide the data and code via the following link: https://anonymous.4open.science/r/HVCR-A9B6/.

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

# A  THE USE OF LARGE LANGUAGE MODELS (LLMS)

In this study, LLMs are used solely for the purpose of polishing the writing and enhancing the clarity of the manuscript. They assist in refining sentence structure, improving grammar, and ensuring overall coherence. It did not contribute to the ideation or content generation of the research. The authors take full responsibility for the integrity of the research and the final content presented in this paper.

# B  SUMMARY OF CAUSALITY-RELATED VIDEO REASONING BENCHMARKS

| Dataset | Realism | | Labeling | | Definition | Evaluation Goal | |
|---|---|---|---|---|---|---|---|
| | Synth | Real | Auto | Human | | Human | Formal |
| IntPhys (Riochet et al., 2018) | ✓ | ✗ | ✓ | ✗ | VOE | ✓ | ✗ |
| IntPhys 2 (Bordes et al., 2025) | ✓ | ✗ | ✓ | ✗ | VOE | ✓ | ✗ |
| CLEVRER (Yi et al., 2020) | ✓ | ✗ | ✓ | ✗ | Heuristic | ✗ | ✓ |
| CLEVRER-Humans (Mao et al., 2022) | ✓ | ✗ | ✗ | ✓ | Human Judgment | ✓ | ✗ |
| CRAFT (Ates et al., 2022) | ✓ | ✗ | ✓ | ✗ | Force Dynamics | ✓ | ✗ |
| MECD (Chen et al., 2024b) | ✗ | ✓ | ✓ | ✗ | Granger Causality | ✗ | ✓ |
| MECD+ (Chen et al., 2025) | ✗ | ✓ | ✗ | ✓ | Granger Causality | ✗ | ✓ |
| CausalVQA (Foss et al., 2025) | ✗ | ✓ | ✓ | ✓ | Physical | ✓ | ✗ |
| CausalStep (Li et al., 2025b) | ✗ | ✓ | ✗ | ✓ | Stepwise Reasoning | ✓ | ✗ |
| VCRBench (Sarkar & Etemad, 2025) | ✗ | ✓ | ✗ | ✓ | Event Dependency | ✓ | ✗ |
| NExT-QA (Xiao et al., 2021) | ✗ | ✓ | ✗ | ✓ | Visible Cause | ✓ | ✗ |
| Finding the Trigger (Le et al., 2025) | ✓ | ✓ | ✓ | ✗ | Causal Abduction | ✗ | ✓ |
| Impossible Videos (Bai et al., 2025b) | ✓ | ✗ | ✗ | ✓ | Anti-Commonsense | ✓ | ✗ |
| GRASP (Jassim et al., 2024) | ✓ | ✗ | ✓ | ✗ | Intuitive Physics | ✓ | ✗ |
| Video-Holmes (Cheng et al., 2025) | ✗ | ✓ | ✗ | ✓ | Narrative Logic | ✓ | ✗ |

Table 4: Comparison of `HVCR` with other causality-related video reasoning benchmarks.

## B.1 HUMAN EVALUATION

---

**Labeling Instruction**

### Goal

The purpose of this project is to investigate human intuition about causal relationships in videos. Your task is to carefully observe the motion and interactions of objects in each video and answer the accompanying questions based on your immediate perceptions.

**Important:** Focus on your intuition rather than whether your answers are "correct" or "incorrect." Respond based on how events appear to you.

### Data

1. *Video Type:* Synthetically generated videos that follow predefined causal structures.

2. *Scene Description:* Objects move or remain stationary on a horizontally infinite tabletop.

3. *Video Length:* 5 seconds (125 frames per video).

4. *Object Properties:*
   - Each object has a unique combination of color, material, and shape.
   - No two objects within the same video share the exact same combination.

5. *Question Types:*
   - Yes/No questions
   - Multiple-choice questions (more than one option may apply)

### Guidelines

1. *Video Viewing Recommendations:* Watch each video multiple times for a comprehensive understanding. Some movements occur rapidly; a single viewing may not capture all relevant details.

2. *Event Types to Observe:*

   (a) Unary Events (single-object events)
   - Motion Status: Determine whether an object is moving or stationary.
   - Motion Transitions: Identify when an object starts or stops moving.
   - Visibility Changes: Note when an object enters or exits the visible area.

   (b) Binary Events (interactions between two objects)
   - Collisions: Identify whether objects collide.
   - Movement Direction Relative to Another Object:
     - Typical scenario: a moving object approaches a stationary object.
     - Critical scenario: a moving object approaches another moving object, creating a potential collision.

3. *Attention to Question Tense:* Pay attention to the tense used in each question (past, present, or continuous) and ensure your responses align with the chronological sequence of events observed in the video.

We wish you an efficient and insightful labeling experience!

---

