# OpenReview forum: "HVCR: Causal Evaluation of Large Multimodal Models in Human-like Video Reasoning"
_ICLR.cc/2026/Conference — ICLR 2026 Conference Withdrawn Submission_

### Official Review · Reviewer_Kcj6 · 2025-10-22

**Soundness:** 3
**Presentation:** 2
**Contribution:** 2
**Rating:** 4
**Confidence:** 5

**Summary:**

This paper proposes the HVCR benchmark for evaluating human-like video causal reasoning in LMMs across three levels (definitional, goal-oriented and representational levels). The HVCR benchmark contains 300 videos (240 synthetic and 60 realistic) and 4,967 causal questions. HVCR is the first video causal reasoning benchmark to systematically integrate  three design levels, jointly consider synthetic and real-world settings, and focus exclusively on pure causal reasoning of LMMs. Finally, several LMMs are evaluated on the HVCR.

**Strengths:**

1)	The HVCR benchmark is built across three levels (definitional, goal-oriented and representational levels).
2)	Both synthetic and real-world videos are included in HVCR.
3)	Detailed description of how to build the HVCR.

**Weaknesses:**

1)	The specific cases of SO, SW, LP, EP, DP and BP in videos and the QA process are missing in the paper.
2)	The main contributions are the HVCR benchmark and some evaluations of the existing LMMs on the HVCR. However, a baseline causal model is not given in this work.
3)	The analysis of Table 2 and 3 lacks deeper insights about these performance and the related metrics.

**Questions:**

See the Weakness.

---

### Official Review · Reviewer_GJ8D · 2025-10-23

**Soundness:** 2
**Presentation:** 3
**Contribution:** 3
**Rating:** 6
**Confidence:** 4

**Summary:**

This paper introduces the HVCR benchmark for evaluating human-like video causal reasoning in LMMs, systematically designed across three levels. The HVCR benchmark contains 300 videos (240 synthetic and 60 realistic) and 4,967 causal questions. These questions span three causal rungs (discovery, intervention, and counterfactual) and eight types, focusing on key aspects of human-like causal reasoning such as causal attribution and responsibility. The authors also conducted an effective evaluation of the performance of current mainstream VLLMs and human participants.

**Strengths:**

1. Video causal reasoning is a very interesting and meaningful research direction. The study presented in this paper is valuable and helps fill certain gaps in the current video causal benchmarks.
2. The paper provides clear explanations of many causal concepts and establishes their correspondences with the video modality.

**Weaknesses:**

1. Although the authors explained many fundamental concepts of causality, I believe several parts may still cause confusion for readers. For example, the sentence *“while the realistic data only considers SW, LP, and BP, as they are more common in real-world situations”* should be further explained. In my view, concepts like SO also frequently occur in real-world scenarios.

2. Although the authors evaluated the performance of many models, I think Gemini-2.5-Pro should also be included in the evaluation. Considering computational cost, a subset of results could be reported. Moreover, since the dataset has no training set, open-source models such as InternVideo could be tested using larger-parameter versions.

3. I believe the QA pair annotation process is very important, and certain necessary quality checks should also be conducted. The paper only mentions *“Based on the causal structure, we generate causal questions using a template-based approach,”* which I think is far from sufficient.

4. Other issues: Some subfigures in Figure 3 have relatively low resolution, and the captions should be more detailed and complete. Although the dataset includes 4k QA pairs, it only contains 300 videos, and there are no open-ended question-answer formats.

5. Some characteristics of the dataset have not been rigorously studied. For instance, how does the model perform when given only the question without the video input? This would clearly reflect the extent of language bias in the dataset.

6. During dataset evaluation, the closed-source models were given 10 frames as input, while the open-source models used 16 frames. I believe this setting is unfair, and I hope the authors can provide a reasonable explanation for this discrepancy.

**Questions:**

See Weakness.

---

### Official Review · Reviewer_c6ko · 2025-10-29

**Soundness:** 1
**Presentation:** 1
**Contribution:** 1
**Rating:** 2
**Confidence:** 4

**Summary:**

This paper claims introduce causal definitions and concepts at three levels. However, these already exist in the filed. Paper then collects 4.9K QA pairs pertaining to 300 videos. These QA span few scenarios. The paper then evaluates a few models on these QA. However, it leaves me with a question, so what? No insights are provided regarding why evaluating across these scenarios and rungs matters. It is not clear to me why we should care about evaluating across these scenarios.

**Strengths:**

Causal understanding and reasoning are important topics. However, this paper fails to do an appropriate survey of existing work; and makes false claims of being first to do a lot of things. However, these things are widely adopted in the causal videoQA field and standard practice. Therefore, originality and novelty of this paper is very minor.

**Weaknesses:**

- Related Work is just one single small paragraph, completely ignoring causal videoQA work and datasets. To name a few: [1,2,3,4,5]
- After such shallow and ignorant discussion of related work, this paper makes a lot of false claims
- What this paper sells as Definitional-level, is already widely followed/adopted and standard practice in causal videoQA
- What this paper sells as Goal-oriented-level has been explored prior work such as [1,2,3,4,5]
- Therefore the novelty is questionable in this work.
- Dataset is quite small in terms of diversity in videos.
- Dataset construction principles are discussed very briefly. For example, how did you ensure high quality of the dataset?
- Human evaluation results seem low, which may indicate low consensus amongst humans over QA---not desirable
- L081-082: Most existing work omits explicit representations of causal relationships (Yi et al., 2020; Foss et al., 2025), while others do (Chen et al., 2024b; 2025; Yi et al., 2020). This sentence structure is wrong. Moreover, authors are using the same reference as doing and not doing.
- Inconsistent number of scenarios are mentioned: sometimes 8 (L026/7), sometimes 7 (L086/7), sometimes 6 (L092/3)
- Sometimes unclear, random stuff is written/mentioned: L215-236 or  L240 "They are also adjusted to enable early modeling of counterfactual knowledge". Like what/how?

- L298: Why CI is learning general rule? L292-300: whole human vs non-human bifurcation seems subjective, not objective.

- L368-370: What does this even mean?

- Experimental result analysis is almost just verbally saying the numbers from tables---insights are missing

- Qualitative results are missing

- After reading the paper, I still have a question: you did this, but so what? Paper does not make its significance/impact clear. It is not clear, why should readers care about the results?


References:

[1] Mao, Jiayuan, et al. "Clevrer-humans: Describing physical and causal events the human way." Advances in Neural Information Processing Systems 35 (2022): 7755-7768.

[2] Parmar, Paritosh, et al. "Causalchaos! dataset for comprehensive causal action question answering over longer causal chains grounded in dynamic visual scenes." Advances in Neural Information Processing Systems 37 (2024): 92769-92802.

[3] J. Li, P. Wei, W. Han and L. Fan, "IntentQA: Context-aware Video Intent Reasoning," in 2023 IEEE/CVF International Conference on Computer Vision (ICCV), Paris, France, 2023, pp. 11929-11940, doi: 10.1109/ICCV51070.2023.01099.

[4] Wu, Bo, et al. "STAR: A Benchmark for Situated Reasoning in Real-World Videos." Thirty-fifth Conference on Neural Information Processing Systems Datasets and Benchmarks Track

[5] Li, Jiangtong, Li Niu, and Liqing Zhang. "From representation to reasoning: Towards both evidence and commonsense reasoning for video question-answering." Proceedings of the IEEE/CVF conference on computer vision and pattern recognition. 2022.

**Questions:**

Please see weaknesses and you may consider them as questions.

---

### Official Review · Reviewer_gej9 · 2025-10-31

**Soundness:** 2
**Presentation:** 2
**Contribution:** 3
**Rating:** 4
**Confidence:** 2

**Summary:**

This work proposes a video causal reasoning benchmark for LMMs that is design on a multi-level principle, definitional level, goal-oriented level and representational level.\
Benchmark contains 300 videos consisting of 240 synthetic and 60 realistic with 4967 causal questions.\
Work focuses on focuses on six causal scenarios with Questions classified three rungs and eight types.\
Authors evaluated on a range of LLMs, illustrating the gaps in LLMs vs Human level performance, demonstrating that the benchmark that is grounded on principles of actual causality can be used to probe and expose the gaps the model have in replicating human-like causal judgments

**Strengths:**

Paper is well-motivated with a clear conceptual framework, that is grounded in actual causality and Pearl’s ladder of causation.\
Benchmark contains both synthetic data and real-work data.\
Authors provide comprehensive evaluations across various models and details of performance across scenarios and rungs.\
The study provides evidence that humans achieve consistent judgments, confirming dataset clarity and alignment with human intuition.\
Clearly defined and labeled scenarios and question classifications can serve to provide valuable insights to causal reasoning ability of models.

**Weaknesses:**

A large portion of the paper is dedicated to covering the preliminaries, and definitions of causality and background concepts. However I think it would have been more beneficial to have details on the dataset construction how these definitions were ensured to be in the question and video as well. i.e for realistic videos that are manually annotated are there any cross-annotator validation or consensus, what criterions are used to questions generated conform to the defined question types and rungs, how are multiple choice generated to prevent biases that can be exploited by language priors

While well defined and clearly motived , empirical section remains largely descriptive without robust statistical analysis or diagnostic ablations that connect failures to specific causal skills.

Authors mention the use of causal graphs and twin networks, but it is not clear to me how these are exploited.
Authors briefly mention the use of "template-based approach", details are lacking here.

**Questions:**

In line 306, causal grpahs and twin networks are provided for each video
In Table 2 and 3, It is clear that the green highlights show the best performance for each row but what do the red highlights represent?

---

### Note · Authors · 2025-11-12

I have read and agree with the venue's withdrawal policy on behalf of myself and my co-authors.